



# Joint inversion of proxy system models to reconstruct paleoenvironmental time series from heterogeneous data

Gabriel J. Bowen[1], Brenden Fisher-Femal[1], Gert-Jan Reichart[2], Appy Sluijs[3], Caroline H. Lear[4]

[1]Department of Geology & Geophysics and Global Change & Sustainability Center, University of Utah, Salt Lake City, UT
84112, USA
[2]NIOZ Royal Netherlands Institute for Sea Research, Den Burg, Texel, Netherlands
[3]Department of Earth Sciences, Faculty of Geosciences, Utrecht University, Utrecht, Netherlands
[4]School of Earth and Ocean Sciences, Cardiff University, Cardiff, UK

*Correspondence to*: Gabriel J. Bowen (gabe.bowen@utah.edu)

**Abstract.** Paleoclimatic and paleoenvironmental reconstructions are fundamentally uncertain because no proxy is a direct record of a single environmental variable of interest; all proxies are indirect and sensitive to multiple forcing factors. One productive approach to reducing proxy uncertainty is the integration of information from multiple proxy systems with complimentary, overlapping sensitivity. Most such analyses are conducted in an ad-hoc fashion, either through qualitative comparison to assess the similarity of single-proxy reconstructions or through step-wise quantitative interpretations where one proxy is used to constrain a variable relevant to the interpretation of a second proxy. Here we propose the integration of multiple proxies via the joint inversion of proxy system and paleoenvironmental time series models in a Bayesian hierarchical framework. The "Joint Proxy Inversion" (JPI) method provides a statistically robust approach to producing self-consistent interpretations of multi-proxy datasets, allowing full and simultaneous assessment of all proxy and model uncertainties to obtain quantitative estimates of past environmental conditions. Other benefits of the method include the ability to use independent information on climate and environmental systems to inform the interpretation of proxy data, to fully leverage information from unevenly- and differently-sampled proxy records, and to obtain refined estimates of proxy model parameters that are conditioned on paleo-archive data. Application of JPI to the marine Mg/Ca and $\delta^{18}O$ proxy systems at two distinct timescales demonstrates many of the key properties, benefits, and sensitivities of the method, and produces new, statistically-grounded reconstructions of Neogene ocean temperature and chemistry from previously published data. We suggest that JPI is a universally applicable method that can be implemented using proxy models of wide-ranging complexity to generate more robust, quantitative understanding of past climatic and environmental change.

## 1 Introduction

Paleoenvironmental reconstructions, including reconstructions of past climate, provide a powerful tool to document the sensitivity of Earth systems to forcing, characterize the range of natural responses associated with different modes of global change, and identify key mechanisms governing these responses. Throughout the vast majority of the planet's history, however,

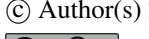



estimates of environmental conditions can only be obtained through proxy reconstructions. The word proxy is derived from the Latin word *procurare*, which in this context means 'to care' or 'to manage'. The measurable physico-chemical quantity in sediments is thus 'managed' into a parameter we want to reconstruct. As implied, the result is an indirect estimate of past environmental conditions, often subject to substantial, sometimes poorly characterized, uncertainty.

5       The simplest proxy reconstructions typically focus on a single environmental variable of interest. Experimental or natural calibration datasets are used to calibrate mathematical relationships between the environmental variable and proxy measure, and these relationships are inverted to obtain quantitative estimates of that variable. Residual variance in the calibration is treated as noise. In reality, however, no proxy exists that is sensitive only to a single paleoenvironmentally-relevant variable, and a large part of the proxy system noise reflects the uncharacterized influence of other environmental and

post-depositional variables. Fossil leaf assemblages, for example, exhibit variability that can be associated with mean annual air temperature, but also may be influenced by many other environmental variables and evolutionary history (Royer et al., 2005;Greenwood et al., 2004). The saturation state of alkenones produced by marine phytoplankton is a sensitive recorder of water temperature, but characteristics of alkenones preserved in marine sediments are also strongly affected by physiological factors, seasonality of production, and selective degradation (Conte et al., 1998;Conte et al., 2006). Even recently emerging

clumped isotope techniques, which are in theory a direct recorder of the temperature of carbonate mineral formation, can be affected by factors such as growth-rate, carbonate system disequilibrium, and poorly constrained, potentially variable offsets between the environment of carbonate formation and more commonly targeted atmospheric temperature conditions (Passey et al., 2010;Affek et al., 2014;Saenger et al., 2012).

      Failure to recognize and consider the sensitivity of proxies to multiple environmental factors leads to two important

problems in traditional proxy interpretations. First, considering only a single environmental variable in our interpretations maximizes the uncertainty in our reconstructions. Uncertainty could be reduced if the influence of other variables is described and constrained. Second, unacknowledged sensitivity to multiple variables creates potential for biased proxy interpretations if variation in these variables is non-random across the reconstruction.

      A productive approach to addressing these issues is the use of proxy system models in the interpretation of proxy data

(Evans et al., 2013). These models represent an attempt to mathematically describe the complex of environmental, physical, and biological factors that control how environmental signals are sampled, recorded, and preserved in proxy measurements. Recent reviews and perspectives are available discussing the concepts underlying proxy system models and different ways that they have been applied to proxy interpretation, ranging from substitution for empirical calibrations in inverse estimation of environmental signals to formal integration within climate model data assimilation schemes (Evans et al., 2013;Dee et al.,

2016). A growing number of proxy system models and modeling systems are being developed (e.g., Tolwinski-Ward et al., 2011;Stoll et al., 2012;Dee et al., 2015), and useful models span a range of complexity from empirically-constrained regressions to mechanistic, theory-based formulations. Key to any such model is accurate representation of uncertainty in each model component, which allows even relatively simple, potentially incomplete models to be used to obtain reconstructions with quantifiable uncertainty bounds.





Reducing the uncertainty of quantitative paleoenvironmental reconstructions, however, further requires adding constraints to proxy interpretations. In situations where two or more proxies share sensitivity to common or complimentary environmental variables, it stands to reason that the information provided by each can be used to refine interpretation of the multi-proxy suite. In practice, a variety of approaches have been used. Commonly, multi-proxy integration has been qualitative

and focused on confirmation: trends reconstructed using one proxy system are cross-checked against a second, providing increased confidence in the reconstruction where the patterns match and further investigation where they don't (e.g., Grauel et al., 2013;Keating-Bitonti et al., 2011;Zachos et al., 2006). In other cases, proxies have been combined quantitatively, but usually in a stepwise fashion: one proxy system is used to reconstruct an environmental variable to which it is sensitive, and those reconstructed values are then used to constrain the interpretation of a second proxy (e.g., Fricke et al., 1998;Lear et al.,

2000). Although it provides a simple strategy to combining complimentary proxy information, this approach does not fully leverage overlapping information that may be contained in multiple systems that respond to common forcing, is not conducive to robust quantification of uncertainty, and requires that both proxies sample coeval paleoenvironmental conditions.

Here we propose a general approach to proxy interpretation that leverages the benefits of proxy models and provides a robust statistical basis for multi-proxy integration. The method, which we call Joint Proxy Inversion (JPI), couples proxy

models with simple environmental time series models representing paleoenvironmental target variables in a Bayesian hierarchical modeling framework (Fig. 1). The hierarchical model is then inverted using Markov Chain Monte Carlo methods (Geman and Geman, 1984) to obtain posterior parameter estimates and paleoenvironmental time series that are conditioned simultaneously on all proxy and calibration data. Similar approaches have been applied in a limited number of cases to conduct large-scale meta-analyses (Tingley and Huybers, 2010;Li et al., 2010;Tingley et al., 2012), but have not found widespread use

in quantitative proxy interpretation. We begin by describing an implementation of JPI for the widely-used foraminiferal Mg/Ca and $\delta^{18}O$ multi-proxy system, and then present results demonstrating many of the merits and challenges of this approach. The examples are not intended to probe a particularly challenging application or to formally test or validate the approach, but rather to illustrate how JPI offers a robust, accessible framework for the types of quantitative proxy data interpretations routinely conducted within the paleoenvironmental research community.

## 25    2 Methods

### 2.1 Data

Proxy and proxy model calibration datasets were compiled from published work (Fig. 1). Estimates from fluid inclusions, calcite veins, large foraminifera, and echinoderm fossils (Dickson, 2002;Coggon et al., 2010;Lowenstein et al., 2001;Evans et al., 2018;Horita et al., 2002) were combined with information on modern seawater Mg/Ca (de Villiers and Nelson, 1999) to

represent variation in seawater Mg/Ca since 80 Ma. For simplicity, and because of the relatively low sensitivity of the other paleoenvironmental variables to seawater Mg/Ca estimates, we use interpreted seawater Mg/Ca estimates given by these authors instead of developing formal models for each Mg/Ca proxy system. Because uncertainty exists in the form of the




partitioning function between seawater and echinoderm carbonate, our dataset includes both the original estimates from Dickson (2002) and the reinterpreted estimates of Hasiuk and Lohmann (2010). The uncertainty associated with each estimate was approximated from the primary publication, and ranged from 0.03 mol/mol for modern seawater to ~0.5 mol/mol for some of the proxy estimates (1 σ, see data and code available at https://github.com/SPATIAL-Lab/JPI_marine).

Foraminiferal Mg/Ca and $\delta^{18}$O data were compiled from three Ocean Drilling Program (ODP) sites: site 806, Ontong Java Plateau (Lear et al., 2015;Lear et al., 2003;Bickert et al., 1993); site 1123, Chatham Rise (Elderfield et al., 2012), and site U1385, Iberian Margin (Birner et al., 2016). All Mg/Ca data are all derived from infaunal foraminifera, which exhibit little to no Mg/Ca sensitivity to changing bottom water saturation state (Elderfield et al., 2010). Data from site 806 constitute a low-resolution record from ~18 Ma to present, with an average sampling resolution of 1 sample per 240 and 180 kyr for Mg/Ca
and $\delta^{18}$O, respectively, prior to 800 ka (sampling for $\delta^{18}$O, in particular, increases several fold thereafter). Mg/Ca measurements were made on *Oridorsalis umbonatus*, and $\delta^{18}$O data represent the benthic genus *Cibicidoides*. For the other two sites, data were extracted for the overlapping period (1.32 – 1.23 Ma) and comprise a set of higher-resolution records (sampling resolution between 1 per 110 and 1 per 1,700 years) spanning two glacial/interglacial cycles. Mg/Ca measurements were made on tests of *Uvigerina* spp at both sites, and $\delta^{18}$O data are from either *Uvigerina* spp (site 1123) or *Cibicidoides wuellerstorfi* (site
U1385). Variance in the foraminiferal data, e.g., due to analytical effects and sample heterogeneity, was not estimated independently but rather treated as a model parameter and conditioned on the calibration and proxy data.

Calibration datasets were compiled to constrain the Mg/Ca and $\delta^{18}$O proxy system models. Mg/Ca calibration data for *O. umbonatus* are from the compilation of Lear et al. (2015), and include both modern core-top samples and samples from Paleocene and Eocene sediments of ODP site 690B. Data from site 690B include an adjustment for differences in cleaning
procedures used for those samples (Lear et al., 2015). For *Uvigerina* spp our reconstructions are based on core-top calibration samples compiled by Elderfield et al. (2010). We also evaluated the now widely-used down-core calibration proposed by Elderfield et al. (2010), which optimizes the foraminiferal Mg/Ca temperature sensitivity to match Holocene to Last Glacial Maximum temperature change inferred from foraminiferal $\delta^{18}$O values and independent constraints on seawater $\delta^{18}$O change. We found that this approach provided relatively weak constraints on the Mg/Ca proxy model parameters and posterior
parameter estimates that were entirely consistent with the stronger constraints obtained from core-top calibration (Fig. S1). Including both calibration datasets in JPI produced results similar to the core-top-only approach; as a result, we exclude the down-core calibration for simplicity, but note that multiple calibration approaches can be integrated and/or evaluated within JPI. Each Mg/Ca datum is accompanied by a bottom water temperature (BWT) estimate based on syntheses of observational data (modern) or $\delta^{18}$O thermometry (paleo), the latter assuming ice-free conditions. We adopt both sets of estimates directly,
applying a normally distributed uncertainty to the BWT values with a standard deviation of 0.2 and 1 °C for the modern and paleo data, respectively, to approximate the different quality of these estimates.

For $\delta^{18}$O we used the compilation of Marchitto et al. (2014) including new and published coretop data for the genera *Cibicidoides* and *Uvigerina* (Keigwin, 1998;Grossman and Ku, 1986;Shackleton, 1974). Estimates of BWT and $\delta^{18}$O of seawater from the original authors were adopted with an estimated uncertainty of 0.2 °C (1 σ) for BWT; as for Mg/Ca we do





not attempt to constrain the uncertainty in the relationship between temperature and $\delta^{18}O$ fractionation between seawater and calcite directly, but treat it as a model parameter.

The age of each pre-modern datum was taken from the primary source. Age uncertainties, where known, can be incorporated in the JPI analysis framework by treating ages as random variables rather than as fixed values and/or including

proxy model components representing processes governing the time-integration of observations. For simplicity, we do not include such a treatment here. In the discussion we note examples where including age uncertainty would produce a more robust analysis.

## 2.2 Proxy models

The proxy system models comprise the 'data model' layer of the hierarchical model, representing how environmental signals

are embedded in the paleo-proxy and proxy calibration data. The models used here are comprised of simple transfer functions relating proxy data to contemporaneous environmental variables, and as such can be considered "sensor models" in the terminology of Evans et al. (2013). The simplest model is that for seawater Mg/Ca proxy data, where, as noted above, we consider the interpreted data directly, giving:

$$MgCa_{swp}(i) \sim N\big[MgCa_{sw}(t_{swp}[i]), \sigma_{swp}(i)\big].$$             Eq. (1)

Here $MgCa_{swp}(i)$ is the $i^{th}$ proxy estimate, $N$ represents the normal distribution, $MgCa_{sw}$ is the paleo-seawater Mg/Ca value, and $t_{swp}$ and $\sigma_{swp}$ are the age estimate and uncertainty associated with a proxy estimate.

We model foraminiferal Mg/Ca ($MgCa_f$, including both calibration and proxy data) as a function of seawater

chemistry and bottom water temperature:

$$MgCa_f(i) \sim N\big[(\alpha_1 + \alpha_2 \times BWT[t_{MgCaf}(i)]) \times MgCa_{sw}(t_{MgCaf}[i])^{\alpha_3}, \tau_{MgCaf}\big],$$     Eq. (2)

where $\alpha_{1\text{-}3}$ and $\tau_{MgCaf}$ are the parameters and precision ($1/\sigma^2$) associated with the transfer function, respectively, and other

parameters are analogous to equation 1. In the absence of theoretical constraints, we assign normally distributed priors to the $\alpha$ parameters based on Bayesian regression of the expression for the mean in equation 2 against the calibration datasets. For *Oridorsalis* we assume paleo-seawater Mg/Ca of 1.5 mol/mol in the Paleocene and Eocene for these initial estimates, and the prior estimates are $\alpha_1 \sim N[1.5, \sigma = 0.1]$, $\alpha_2 \sim N[0.1, \sigma = 0.01]$, and $\alpha_3 \sim N[-0.02, \sigma = 0.03]$. For *Uvigerina* these distributions are $\alpha_1 \sim N[1.02, \sigma = 0.1]$ and $\alpha_2 \sim N[0.07, \sigma = 0.01]$, and the prior estimated for $\alpha_3$ from the *Oridorsalis* data set was used

because no calibration data were available representing non-modern $MgCa_{sw}$. For both genera, the prior estimate on the precision of the foraminiferal Mg/Ca model, $\tau_{MgCaf}$, is the gamma distribution $\Gamma[\text{shape} = 2, \text{rate} = 1/30]$, which approximates the precision of the independent regressions.



Foraminiferal calibration and proxy $\delta^{18}O$ values ($\delta^{18}O_f$) are modeled similarly, with:

$$\delta^{18}O_f(i) \sim N\left[\delta^{18}O_{sw}\left(t_{\delta18Of}[i]\right) + \beta_1 + \beta_2 BWT\left[t_{\delta18Of}(i)\right] + \beta_3 BWT\left[t_{\delta18Of}(i)\right]^2, \tau_{\delta18Of}(i)\right]. \quad \text{Eq. (3)}$$

Here $\delta^{18}O_{sw}$ is the modeled seawater isotope composition and $\beta_{1-3}$ are the transfer function coefficients. In this analysis we treat the scale conversion factor between the SMOW and PDB reference scales (Shackleton, 1974) as implicit in the transfer function intercept term ($\beta_1$), which is relevant only in comparing our posterior parameter estimates to other work. Prior estimates of the model parameters were obtained and specified as for Mg/Ca; these are $\beta_1 \sim N[3.32, \sigma = 0.02]$, $\beta_2 \sim N[-0.237, \sigma = 0.01]$, $\beta_3 \sim N[0.001, \sigma = 0.0005]$ for *Cibicidoides* and $\beta_1 \sim N[4.05, \sigma = 0.06]$, $\beta_2 \sim N[-0.215, \sigma = 0.02]$, $\beta_3 \sim N[-0.001, \sigma = 0.001]$ for

*Uvigerina*. Because the amplitude of high-frequency (i.e. below the resolution of our model) $\delta^{18}O_{sw}$ variance in the record from site 806 increased substantially with the onset of modern, 100 kyr glacial cycles, we modeled $\tau_{\delta18Of}(i)$ separately for proxy data younger than 800 ka (prior on $\tau_{\delta18Of} \sim \Gamma[6, 1]$) and for all other proxy and calibration data ($\Gamma[3, 1/30]$). The former estimate is based on the observed proxy variance since 800 ka, whereas the latter approximates the precision of the calibration relationships.

**2.3 Environmental models**

Although not treated as such in most reconstructions, paleoenvironmental conditions are autocorrelated in time, meaning that each proxy observation provides information about conditions not just at a single point in time but across a segment of time. To reflect this, we model paleoenvironmental variables as time series using a correlated random walk model. This parameterization is desirable in that it is minimally prescriptive (i.e. no preferred state or pattern of change is proscribed) but

allows incorporation of constraints on (and extraction of inference about) two basic characteristics of the paleoenvironmental system – namely its rate and directedness of change. The environmental models represent the "process model" layer of the Bayesian hierarchical model.

The correlated random walk for variable $Y$ is expressed as:

$$Y(t) = Y(t-1) + \epsilon_Y(t), \hspace{6cm} \text{Eq. (4)}$$

where:

$$\epsilon_Y(t) = N[\phi_Y \times \epsilon_Y(t-1), \tau_Y]. \hspace{5cm} \text{Eq. (5)}$$


In short, the variable follows a random walk in which the next value in the time series is a function only of the current value and a normally distributed error term $\epsilon_Y$, which has a temporal autocorrelation of $\phi_Y$ and precision $\tau_y$. This gives three





independent parameters, $\varphi_Y$, $\tau_y$, and an initial value of $Y$ at the beginning of the time series. We do not explicitly model the covariance among environmental variables, but let this emerge from the data.

For seawater Mg/Ca, which is thought to evolve gradually (the oceanic residence times of Mg and Ca are 13 Ma and 1 Ma, respectively) in response to long-term tectonic and biogeochemical forcing (Wilkinson and Algeo, 1989), we simulate
the time series at 1 Myr steps from 80 Ma to present. Although the foraminiferal proxy data used here span only the interval from ~18 Ma to present, extending the seawater model over this longer temporal domain was necessary in order to generate a stable time series, conditioned on sparse seawater Mg/Ca proxy data, that spanned both the proxy records and the Paleogene-aged Mg/Ca proxy calibration data. Given that the modeled quantity is a ratio, we treat the error term in this time series model as a proportion, such that the change in $MgCa_{sw}$ between two time steps is $MgCa_{sw}(t-1) * \epsilon_{MgCasw}$. We adopt priors that imply
relatively slow change and strong temporal trends ($\varphi_{MgCasw}$ is given by a uniform distribution, U[0.9, 1]; $\tau_{MgCasw} \sim \Gamma[100, 0.01]$). We use a weak prior on the initial state of $MgCa_{sw}$ at 80 Ma, U[1, 3], consistent with independent interpretations of Cretaceous proxy data (Coggon et al., 2010).

We select the bounds, resolution, and prior distributions for the bottom water temperature and $\delta^{18}O$ time series models based on the properties of each record. For site 806 we use a time step of 50 kyr from 18 Ma to present, adequate to allow the
time series model to adapt across the range of supra-orbital timescales represented in the sample distribution. Prior estimates of the error term parameters were chosen to allow sampling across a range of weak to moderate autocorrelation states and error variances that were consistent with first-order interpretations of the proxy data ($\varphi \sim$ U[0, 0.4] for both proxies; $\tau_{BWT} \sim \Gamma[20, 2]$; $\tau_{\delta18Osw} \sim \Gamma[10, 0.2]$). We use weakly informative uniform priors for initial values at 18 Ma ($BWT(-18) \sim$ U[3, 8], $\delta^{18}O_{sw}(-18) \sim$ U[-1, 1]). For the higher-resolution Pleistocene records, we bound the models between 1.32 and 1.235 Ma and adopt a time
step of 1 kyr, accommodating orbital time-scale changes in the paleoenvironmental variables. We adopt the same prior distributions for $\tau_{BWT}$ and $\tau_{\delta18Osw}$ as in the long-term model, but use a broader prior on $\varphi$ (U[0, 0.8] for both environmental variables) based on the expectation that temporal autocorrelation in temperature and seawater $\delta^{18}O$ trends may be stronger at timescales of 1 kyr than at 50 kyr.

## 2.4 Model inversion

The model structure described above was coded in the BUGS (Bayesian inference Using Gibbs Sampling) language (Lunn et al., 2012) and Markov Chain Monte Carlo was used to generate samples from the posterior distribution of all model parameters conditioned on the proxy and calibration datasets. The analysis was implemented in R version 3.5.1 (R Core Team, 2018) using the rjags (Plummer, 2018) and R2jags (Su and Yajima, 2015) packages. Three chains were run in parallel. Convergence was assessed visually via trace plots and with reference to the Gelman and Rubin convergence factor (Rhat; Gelman and Rubin,
1992) and effective sample sizes reported by rjags.

For the site 806 analysis, chains were run to a length of $1.5e^6$ samples with a burn-in period of $10e^5$ samples and thinning to retain a total of 5,000 posterior samples. All parameters showed strong convergence (Rhat << 1.05, effective sample size > 3,500) with the exception of some parts of the seawater Mg/Ca time series and the initialization period of the $BWT$ and




$\delta^{18}O_{sw}$ time series (i.e. prior to the first proxy observation). The long run and burn-in periods were dictated by the $MgCa_{sw}$ time series values, which exhibited very strong autocorrelation as a result of their 'stiff' time series behavior and weak data constraints. Qualitative assessment showed no perceptible covariance between seawater Mg/Ca and other parameters in the posterior samples, nor was the posterior distribution obtained from this inversion substantially different from one produced by

inverting the Mg/Ca proxy model alone (which was run to an effective sample size >4,000 beyond the initialization period); as a result, we do not believe the weaker sampling from the $MgCa_{sw}$ posterior has a significant impact on our results or interpretations. The entire analysis took approximately 22 hours running on three cores of a Windows desktop computer.

For the Pleistocene data we conducted three different analyses, the first two inverting data from each site independently and the third inverting both records together. Because of the short time interval covered by these analyses we

did not model the seawater Mg/Ca explicitly, but estimated paleo-seawater Mg/Ca values, where needed, from the posterior distributions of an independent inversion of the seawater Mg/Ca proxy data. Chains were run to $5e^5$ and $7.5e^5$ samples for the single- and multi-site analyses, respectively, using a burn in period of $1e^4$ samples and thinning to retain 5,000 posterior samples. All parameters showed strong convergence (Rhat << 1.05) and effective samples sizes were >4,000 for most parameters and >2,000 for all parameters excluding the initialization period of the time series (i.e. prior to the first observation).

Total analysis time ranged from <1 hour (site 1123) to ~4 hours running three chains in parallel.

## 3 Results and Discussion

### 3.1 JPI paleoenvironmental reconstructions

The paleoenvironmental reconstructions obtained by applying JPI to the site 806 data are similar, to first order, to the reconstructions from Lear et al. (2015; hereafter L15) on which our analysis was modeled (Figs. 2 and 3). Our estimates of

seawater Mg/Ca match those obtained by L15 using polynomial curve-fitting throughout most of the common period of analysis (Fig. 2). Prior to 40 Ma our estimates diverge somewhat, reflecting the incorporating additional data in our analysis, but this difference does not impact other interpretations given that L15 did not use the curve-fit estimates from this part of the record in their analysis. Our reconstruction shows strong support for ~2 °C of bottom-water warming at site 806 during the mid-Miocene Climatic Optimum (centered here on ~15.5 Ma), and although abrupt cooling followed this event, water

temperatures warmed again by ~1 °C into the late Miocene (Fig. 3). A strong and sustained multi-Myr cooling trend began at the site just prior to 5 Ma and persisted throughout the remainder of the record. Our median temperature estimates are most similar to those obtained by L15 using their "NBB" calibrations, which was based on the same compilation of calibration data used here. 95% credible intervals estimated from JPI average 2.8 °C and 0.8 ‰, which is similar to but slightly larger than the uncertainty bounds provided by L15 based on iterative estimation using different calibration functions. The width of the JPI

CIs varies subtly across the time series, with somewhat narrower intervals during periods of dense sampling, e.g., in the late Pleistocene.



JPI paleoenvironmental time series for the single- and multi-site analysis of the Pleistocene data were nearly identical, with slightly broader credible intervals for both parameters (BWT and $\delta^{18}O_{sw}$) and sites in the single-site analyses (Figs. S2 and S3). The multi-site analysis showed coherent and slightly phase-shifted patterns of BWT variation across glacial-interglacial cycles at the two sites, with the amplitude of variation being approximately twice as high and median BWT estimates 2 to 5 °C warmer at U1385 (Fig. 4a). Reconstructed $\delta^{18}O_{sw}$ values show greater glacial-scale variability at site 1123, with abrupt decreases of ~0.5‰ accompanying both glacial terminations (Fig. 4b). In contrast, the seawater $\delta^{18}O$ time series reconstructed for site U1385 shows little response to the termination at ~1.295 Ma and exhibits high-frequency variability not seen at 1123. Both reconstructions are similar in nature to those provided by the original authors. Absolute temperatures and $\delta^{18}O_{sw}$ values match well if the published reconstructions are adjusted using the Mg/Ca proxy sensitivity inferred here (0.068 mmol/mol per degree; Fig. 4); the Elderfield et al. (2010) calibration used by the original authors offsets the warmer site U1385 temperatures from JPI results by as much as ca. -2 °C (Figs. S2 and S3). Neither of these studies presents quantitative uncertainty bounds on individual paleotemperature or $\delta^{18}O_{sw}$ estimates, but both provide estimates of average uncertainty based on propagation of errors. The average width of our 95% CIs is actually somewhat narrower than the 2σ values from the original papers, and the JPI CIs are notably narrower for the U1385 record (2.3 °C, 0.6‰) than for 1123 (2.9 °C, 0.7‰; all estimates from the multi-site analysis).

### 3.2 Time series properties

One visually striking difference between the JPI and L15 reconstructions is the higher BWT and $\delta^{18}O_{sw}$ variability implied by L15 (Fig. 3). As is common in traditional proxy interpretations, the L15 paleoenvironmental record treats each individual proxy observation as an estimate of an independent environmental state, giving a reconstruction centered on 'best estimates' derived from each data point. In reality, however, the environmental states giving rise to the proxy data are not independent if autocorrelation exists at the resolution at which the time series is sampled. For BWT and $\delta^{18}O_{sw}$ this is true over a broad spectrum of temporal resolutions including those considered here; e.g., values of these parameters are known to vary systematically over millions of years due to long-term fluctuations in Neogene climate and ice volume (Zachos et al., 2001;Raymo and Ruddiman, 1992) and over tens to hundreds of thousands of years due to orbital forcing (Imbrie et al., 1984;Shackleton, 2000). This is often implicitly acknowledged in the presentation of traditional proxy reconstructions by including a smoothed representation of the record, obtained using a (usually somewhat arbitrary) filter (e.g., Elderfield et al., 2012).

JPI, in contrast, explicitly considers temporal autocorrelation of the underlying environmental variables, treating each proxy observation as a sample arising from one or more underlying, autocorrelated environmental time series. The properties of the time series themselves, rather than being assumed, are estimated using the proxy models and the data, meaning that the record produced is optimized to reflect the actual information content of the data. For very certain proxy models or densely distributed data that record high-frequency variability, the reconstructed time series will express short-term changes in the environment, whereas reconstructions based on uncertain models or smooth or sparsely-sampled data will tend toward greater





smoothing and reflect the actual information content of the proxies with respect to the longer-term evolution of the mean state of the system. This is nicely illustrated by comparison of JPI $\delta^{18}O_{sw}$ reconstructions for sites 1123 and U1385: the sample density of the U1385 proxy record is approximately 15 times greater, and the resultant time series reconstruction expresses much stronger variability at millennial timescales (Fig. 4b).

Another advantage of embedding time series models in JPI is that it offers an explicit framework for integration of differently-sampled proxy records. In most of the studies reviewed here foraminiferal $\delta^{18}O$ values are more densely sampled than Mg/Ca. In a traditional, piece-wise interpretation of these proxy data, $\delta^{18}O_{sw}$ can only be estimated if paired oxygen and Mg/Ca data are available for a given core level. Thus, if Mg/Ca data are missing at a level either this value must be estimated, usually through linear interpolation, or the foraminiferal $\delta^{18}O$ data excluded from the analysis. JPI eliminates the need to
exclude or selectively interpolate data by linking all proxy measurements to a common set of continuous time series. The temporal interpolation required to integrate data sampled at different times is conducted for each environmental variable (which are in reality the quantities that are related in time), rather than for the proxy values themselves, as an explicit component of the analysis. One note of caution is warranted here: potential for artefacts to emerge from the integration of datasets with very different sampling densities remains. For example, the high-frequency variability in estimated seawater $\delta^{18}O$ at site U1385
(Fig. 4b) stems from high-frequency variance in the densely-sampled $\delta^{18}O_f$ record at this site, but without $MgCa_f$ at similar resolution it is impossible to determine whether the isotopic proxy record variance truly reflects millennial-scale changes in seawater $\delta^{18}O$ or instead is driven by un-documented, high-frequency BWT variation.

       A final outgrowth of the integration of proxy system and paleoenvironmental time series models via JPI is that the method provides quantitative uncertainty bounds that are linked to and reflect the stratigraphic distribution and density of
proxy information. Because environmental parameters are modeled as continuous time series, estimates of central tendency and dispersion (e.g., credible intervals) are obtained throughout the reconstruction period. For time steps in which no observational data are available, the dispersion of posterior estimates increases consistent with the properties of the time series model (e.g., between ~55 and 75 Ma in the seawater Mg/Ca model; Fig. 2), providing quantitative estimates of the constraints provided by the data within these intervals. Moreover, because temporal autocorrelation of the environmental variables is
considered, densely sampled data, even where samples are taken at different stratigraphic levels, place additive constraints on the reconstructed value of the environmental state. As a result, credible intervals in the posterior distribution adjust to reflect both the density and the strength of the proxy constraints. The result can be seen, for example, in the broader 95% CIs for the sparsely-sampled portion of the site 806 record between ~7 and 10 Ma (Fig. 3) or in the contrasting width of the CIs for the two Pleistocene sites (Fig. 4).

**3.3 Model properties**

Bayesian inversion has previously been used to estimate proxy model parameter values in situations where these are poorly constrained (Tolwinski-Ward et al., 2013), and the joint inversion of proxy and environmental time series models performed in JPI can similarly be used to provide constraints on parameter values for all model components. Because the proxy system



models used here are simple statistical formulations, and the calibration data themselves used to generate prior estimates on model parameters, the mean posterior estimates are generally quite similar to the priors (Fig. 5). The only notable exception is $\beta_3$ the second-order parameter in the $\delta^{18}O_f$ model, for which the posterior mean is shifted subtly toward zero (Fig. 5g). Our prior estimates of parameter variance were slightly inflated to ensure that we did not over-constrain these values, and the

posteriors show sharpening of the distributions for most parameters. Posterior estimates for proxy model precision (or variance), however, are much more strongly constrained than those obtained from independent estimation using calibration data only (Figs. 5d and h).

These refinements reflect a combination of the constraints offered by the calibration and down-core proxy data. Although at first consideration the relevance of the latter to calibrating proxy model parameters might not be apparent, in fact

the proxy model must not only be consistent with the calibration data but also explain the observed proxy data given the 'true' environmental conditions. As a result, for a given set of proxy data and environmental time series model properties only a subset of proxy model parameter values will be plausible. Consider, for example, the proxy model precision parameter. In our model construction, this value explains the "noise" both within the model calibration dataset and the proxy record, each of which can arise from a similar ensemble of factors (e.g., temporal variation in the environment at time scales below the time

series model time step, biological or random variation in the environment-proxy relationship). Our analysis suggests that before the mid-Pleistocene transition, the proxy model variance implied by the full JPI inversion is similar to that estimated from the calibration data alone (solid curves in Figs. 5d and h), with slightly higher $\delta^{18}O$ and lower Mg/Ca variance implied by the full analysis. The site 806 $\delta^{18}O_f$ record, however, is much more densely sampled after 800 ka, and the combination of higher $\delta^{18}O_{sw}$ variability and dense sampling that more strongly records this variability requires a much higher proxy model variance (dashed

lines in Fig. 5h). The proxy calibration data offer no constraints on this value, rather the JPI posterior estimates the parameter value to reconcile the environmental time series (representing the longer-term evolution of the mean system state) with the variance expressed in the proxy observations.

Because the JPI analysis involves sampling of all model parameters simultaneously, it also can identify and account for correlation among parameters. The proxy model parameter estimates for site 806 provide a clear example (Fig. 6). The

posterior distributions show strong correlation between the seawater Mg/Ca sensitivity term ($\alpha_3$) and both the intercept and sensitivity terms ($\alpha_1$ and $\alpha_2$) in the $MgCa_f$ model and between the first- and second-order terms ($\beta_2$ and $\beta_3$) in the $\delta^{18}O_f$ model. This is not at all surprising: in all cases these terms are interactive and for a given estimate of the model calibration a change in one will generally be offset by a change in the other. Accounting for this covariance is important in assessing the uncertainty of proxy reconstructions, however, and may in part account for the more optimistic uncertainty estimates obtained here relative

to those based on propagation of errors assuming independence of parameters, in that the latter approach will 'double-count' uncertainty associated with correlated parameters.

JPI also provides posterior estimates on the environmental time series model parameters, and these distributions can provide information complimentary to the reconstructed time series themselves. Comparing prior and posterior estimates at all three study sites (Fig. 7), the analysis provides strong posterior constraints on the error autocorrelation (i.e. directedness of



change) and the error variance (i.e. magnitude of change between time steps) for $\delta^{18}O_{sw}$, but posterior estimates of $BWT$ error variance are only subtly different from the priors (middle column). Interestingly, the error variance estimates are quite similar for both environmental variables at all sites despite the ~2 order of magnitude difference in the resolution of the time series models and data density, suggesting scale-independence of short-term rates of change in these systems.

5     In contrast, the error autocorrelation term, which reflects the directedness of environmental change across multiple model time steps, shows substantial variation among the data sets (Fig. 7, left column). The lowest posterior values were obtained for the long record at site 806, consistent with the assumption that inertia would be weaker for these variables at the longer time scales (i.e. 50 kyr time steps) reflected in this analysis. Across all scales, posterior distributions for autocorrelation were skewed lower for $\delta^{18}O_{sw}$ than for $BWT$. Although this may in part reflect the greater expression of short-term variance in more densely sampled $\delta^{18}O_f$ records, the result holds at site 1123, where the sample distributions for $MgCa_f$ and $\delta^{18}O_f$ are identical, implying that changes in $\delta^{18}O_{sw}$ are generally more chaotic than those of $BWT$. The strongest error autocorrelation is inferred for $BWT$ at site U1385, where the data strongly support highly coherent, high-amplitude cyclic variation in $BWT$ across the two glacial cycles sampled. In contrast, $\delta^{18}O_{sw}$ variation estimated at this site is only weakly directional and features strong, chaotic, millennial-scale variability, reflected in a much lower posterior estimate for error autocorrelation (Fig. 7d).

## 3.4 Derivative analyses

In this final section, we explore additional examples of how JPI results might be used to support inference or hypothesis testing in paleoenvironmental reconstruction. JPI provides a sound basis for testing hypotheses of change within or between proxy records. As with the evaluation of reconstruction uncertainty, the important concept here is that parameter values within individual posterior samples are not independent, but instead reflect the covariance of parameters as constrained by the data and models used. Consider the case where we want to assess the magnitude of change in site 806 bottom water temperature relative to the modern (core top) value. Traditionally, we might take a central estimate of the modern value, e.g., the median shown by the left terminus of the red line in Fig. 8a, and compare it with the reconstructed distribution of values at one or more points in the past to ask whether it is or is not consistent with that distribution. This implicitly assumes that the true environmental parameter values (for modern and the past) are independent of each other, and gives the set of probabilities shown in dotted blue in Fig. 8a. In reality, however, the modern and paleo-$BWT$ values are not independent, as discussed above.

     We can account for this by framing the analysis in terms of change within individual posterior samples (Fig. 8a, solid blue line). The resulting estimates show interesting, if subtle, contrasts with the traditional approach. At short time lags (less than ~400 kyr) the within-sample comparison actually implies somewhat higher probability of significant change. This reflects the influence of error autocorrelation in the time series model: within an individual posterior sample, directional change is likely to persist over multiple time steps, meaning that the 'signal to noise ratio' over short periods is higher if estimated based on within-sample vs. between-sample change. Beyond this time frame, however, the relationship between methods inverts, and the traditional method assuming independence gives exaggerated estimates of the significance of change. Beyond the scale



of significant time series error autocorrelation, the variance of change estimated from the within-sample comparison is actually slightly greater than that estimated between samples, reflecting the fact that some possible BWT trajectories within the posterior 'wander' across the distribution of possible values over time, increasing the dispersion of the change estimates. The net result is that in this case, using a one-sided 95% credible interval threshold (equivalent to $p$=0.05), one would estimate that

site 806 bottom water temperatures diverged from modern some 100 – 200 kyr earlier using the traditional approach than with the more appropriate within-sample analysis.

      Another example involves cross-site comparison. Here, we similarly ask whether seawater $\delta^{18}$O values were different at sites 1123 and U1385 throughout the period of study based on comparisons of the posteriors from the multi-site analysis or the two single-site JPI analyses (Fig. 8b). The assessment that assumes independence of estimates at the two sites (the latter

one) consistently under-estimates the significance of the difference between the sites. This can be explained intuitively in terms of the impact of other model parameters on posterior estimates of $\delta^{18}O_{sw}$ values at both sites. In a given sample from the posterior of the multi-site analysis, if one of the $\delta^{18}O_f$ proxy system model parameters deviates from the central estimate, for example, it will similarly impact the seawater isotope reconstructions at both sites. As a result, the variance of the between-site differences is reduced in the comparison based on the multi-site analysis, producing stronger results in the post-hoc tests of difference

of difference. In this example the choice of approach would have little impact on inferences drawn based on the 95% credible interval, but at the 99% level several parts of the time series would be considered different using the multi-site comparison and not different with the traditional approach (Fig. 8b). Including factors contributing to age model uncertainty for individual records would further improve JPI-based interpretations of this type.

      Finally, because JPI results provide integrated, self-consistent estimates of multiple environmental variables, it can

be used to identify and characterize multivariate modes of environmental change in Earth's past. Results from the site 806 analysis, for example, demonstrate non-linear coupling between changes in $BWT$ and $\delta^{18}O_{sw}$ since the mid-Miocene (Fig. 9). These patterns, including limited coupling between $\delta^{18}O_{sw}$ and BWT change prior to ~5 Ma and strong bottom water cooling accompanied by a modest $\delta^{18}O_{sw}$ decrease into the Pleistocene, were previously noted by L15. What is apparent here, however, is the suggestion that the system transitioned between at least three semi-stable states during this time. Jumps between a mid-

Miocene warm, low-$\delta^{18}O_{sw}$ state, late Miocene warm, high-$\delta^{18}O_{sw}$ state, and Plio-Pleistocene cool state were in each case relatively abrupt, with the system spending the majority of the reconstruction period within, rather than between, states. Patterns of short-term correlation between $BWT$ and $\delta^{18}O_{sw}$ appear to vary among these states, as well, with positive correlation between these variables dominating the first two states and the classical negative correlation characteristic of coupled temperature and ice volume changes only expressed during the final one (Fig. 9, dots).

**4 Conclusion**

Traditional approaches to proxy interpretation suffer from broad and poorly characterized uncertainty and potential biases related to the sensitivity of proxies to multiple environmental factors (Sweeney et al., 2018). Proxy system modeling and multi-

proxy reconstruction provide partial solutions to these issues, but a robust, accessible framework for integrating these two approaches in the development of paleoenvironmental reconstructions is also needed. We suggest that Bayesian hierarchical models that leverage simple time series representations of paleoenvironmental conditions offer such a framework. This approach is broadly generalizable to any set of proxies for which appropriate forward models can be written. It confers many

of the advantages of more complex data assimilation methods that leverage Earth system models (Evans et al., 2013), while remaining independent of the assumptions embedded in these models and flexible enough to be applied over a wide range of systems and time scales. As with any statistically-based analysis, JPI results are model-dependent: they provide a basis for interpreting data in the context of a specific model and its assumptions, and this dependence should be acknowledged and considered in the presentation and interpretation of results.

10        Our illustration of the method based on the coupled Mg/Ca and $\delta^{18}$O systems in benthic foraminifera demonstrates the flexibility of JPI through applications to two contrasting time scales and both single- and multi-site proxy records. Despite the simplicity of this system and the proxy models used, the example illustrates how JPI can be applied to widely used proxy systems to give improved characterization of uncertainty, explicit estimates of the properties of paleoenvironmental systems, and refined proxy model calibrations. Implementations similar to those demonstrated here could easily and immediately

become standard practice in the interpretation of many paleoenvironmental proxy data. As the underlying proxy system models mature, JPI-based interpretations can be revised and refined to incorporate new understanding and/or leverage additional proxy types, minimizing, but also accurately representing, bias and uncertainty in our paleoenvironmental reconstructions.

**Data and code availability**

All data and code used to conduct the analyses and create figures reported in this manuscript are available at

https://github.com/SPATIAL-Lab/JPI_marine.

**Author contribution**

GJB conceived of, designed, and conducted the analyses, with support from BFF, AS, and G-JR. CHL provided access to data and advice on application of the Mg/Ca paleo-thermometer. GJB wrote the manuscript with input from all coauthors.

**Competing interests**

The authors declare that they have no conflict of interest.



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





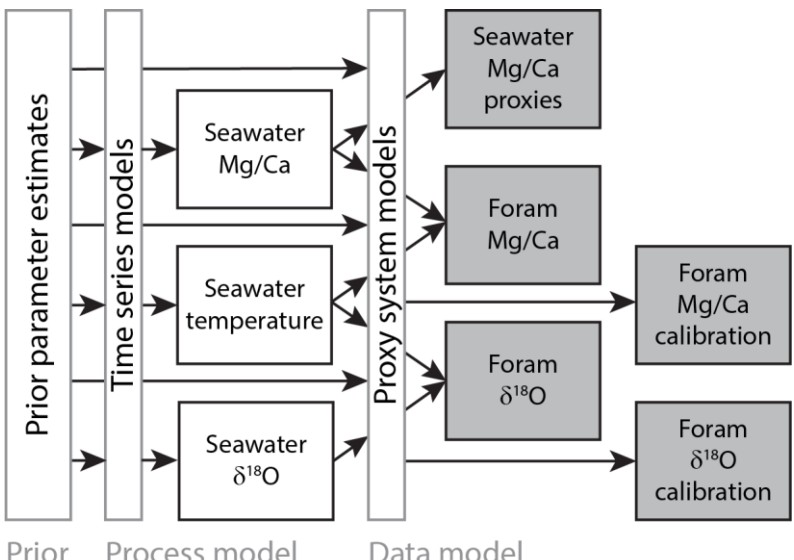

**Figure 1: Schematic representation of the implementation of JPI for the coupled Mg/Ca and $\delta^{18}$O proxy systems. Grey-outlined boxes and text represent the three components of the Bayesian hierarchical model. Markov Chain Monte Carlo sampling is used to 'explore' the prior parameter space and develop a statistically representative posterior sample of the parameters and paleoenvironmental time series that are consistent with all paleo proxy and proxy calibration data (grey-filled boxes).**





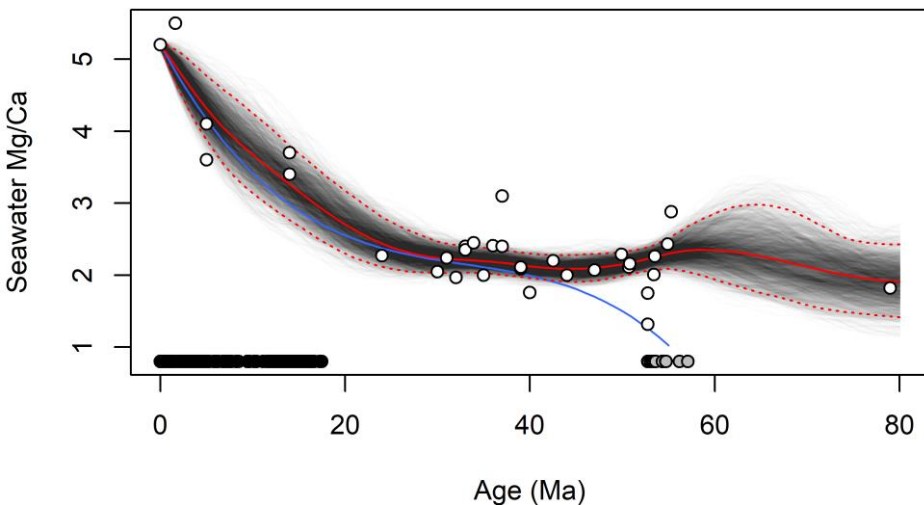

**Figure 2: Reconstructed seawater Mg/Ca from 80 Ma to present. Black lines show individual draws from the posterior distribution for each time series; red lines show the median (solid) and 95% credible intervals (dotted). White-filled circles show individual proxy estimates (Dickson, 2002;Coggon et al., 2010;Lowenstein et al., 2001;Evans et al., 2018;Horita et al., 2002;de Villiers and Nelson, 1999), black and grey symbols at the bottom of the panel show the distribution of the foraminiferal Mg/Ca proxy data and Paleogene proxy calibration data, respectively, in time. The blue line is the curve-fit estimate of seawater Mg/Ca of Lear et al. (2015).**



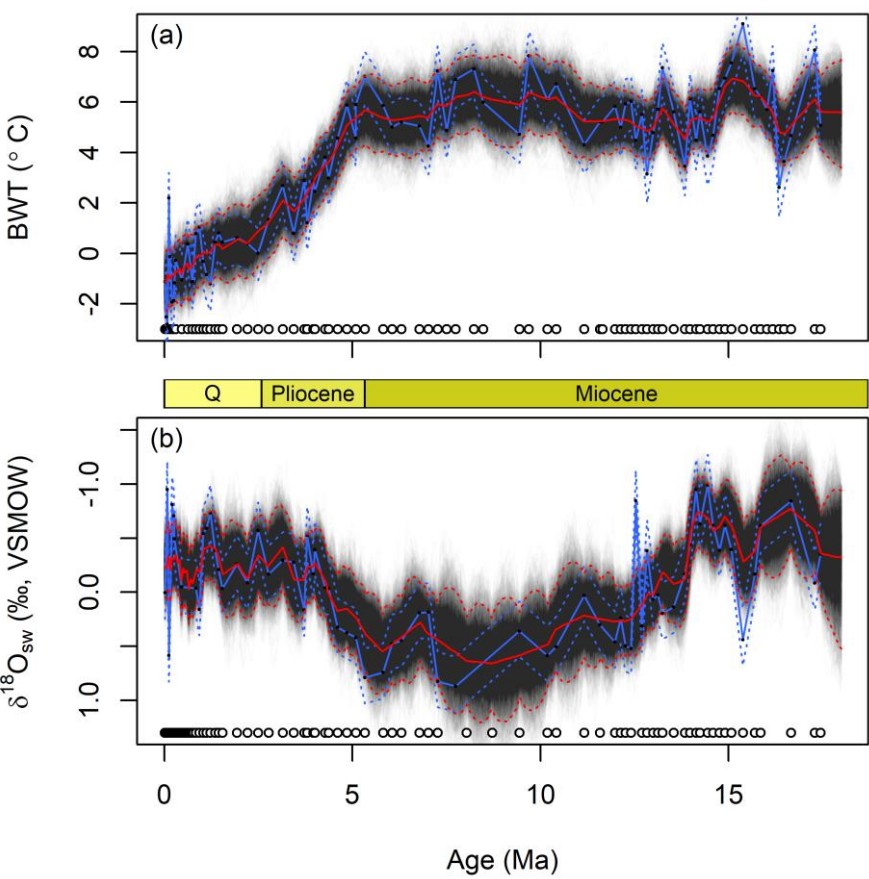

**Figure 3: Reconstructed bottom water temperature (a) and seawater δ¹⁸O values since 18 Ma (b). Lines as in Fig. 2. Circles show the distribution of foram Mg/Ca (a) and δ¹⁸O (b) data in time. Blue lines are the best estimate (solid) and uncertainty envelope (dashed) of the original Lear et al. (2015) interpretation of these data, using their linear "NS-LBB" calibration data set. Q = Quaternary.**





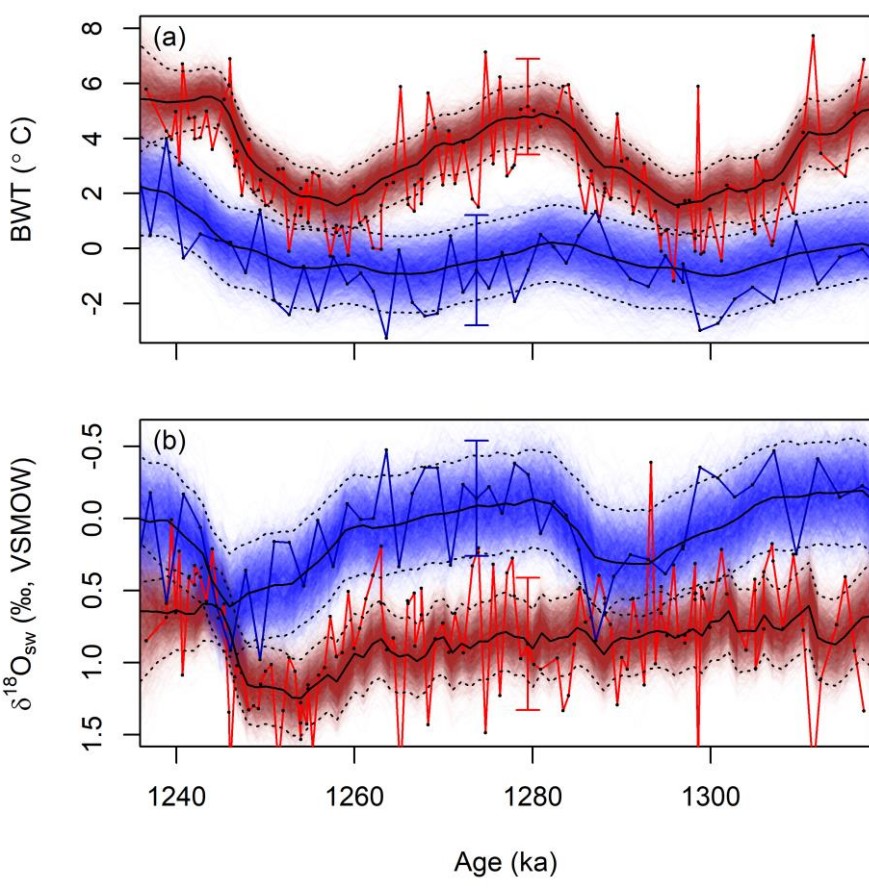

**Figure 4: Reconstructed bottom water temperature (a) and δ¹⁸O values (b) for sites 1123 (blue) and U1385 (red) based on simultaneous JPI of proxy data from both sites. Symbols as in Fig. 2. Solid red and blue lines show the interpretation of these records as by the original authors (Birner et al., 2016;Elderfield et al., 2012) recalculated using the foraminiferal Mg/Ca temperature 5 sensitivity inferred here. Uncertainty estimates from the original authors (2σ) are shown as error bars.**





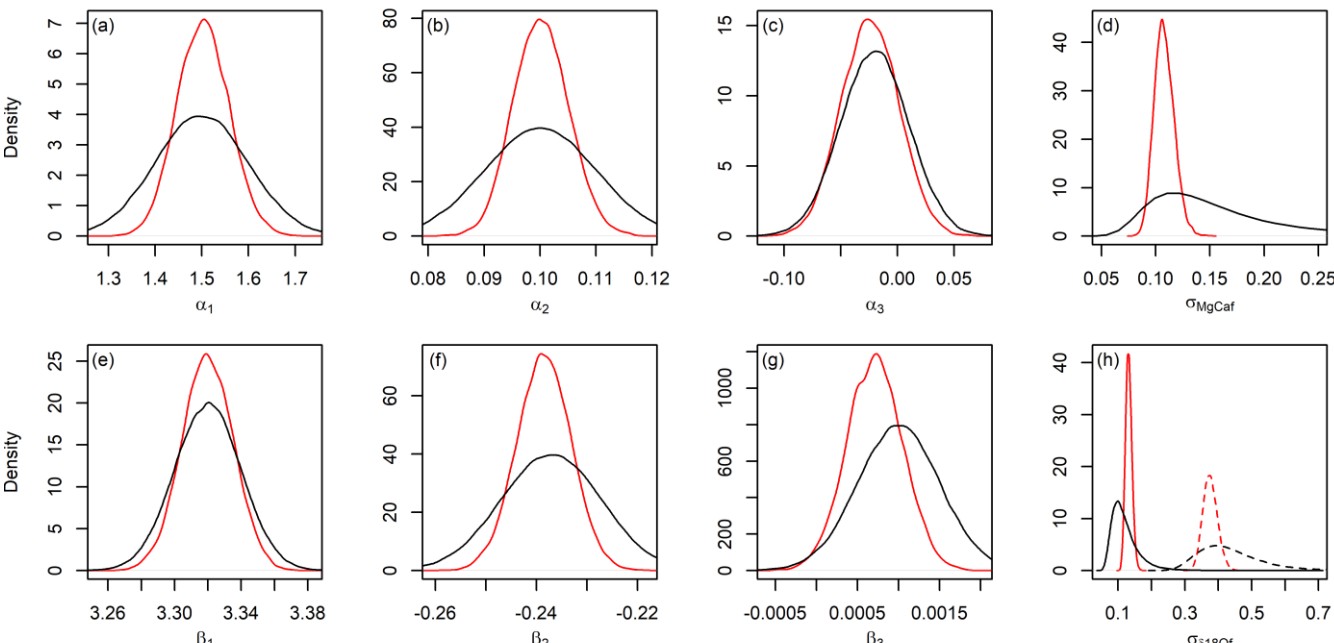

**Figure 5: Prior (black) and posterior (red) distributions for foram Mg/Ca (a-d) and $\delta^{18}$O (e-h) proxy model parameters (ref. equations 2 and 3, respectively). Solid and dashed lines in panel H show standard deviations of the calibration relationship prior to and following the 800 ka transition, respectively.**





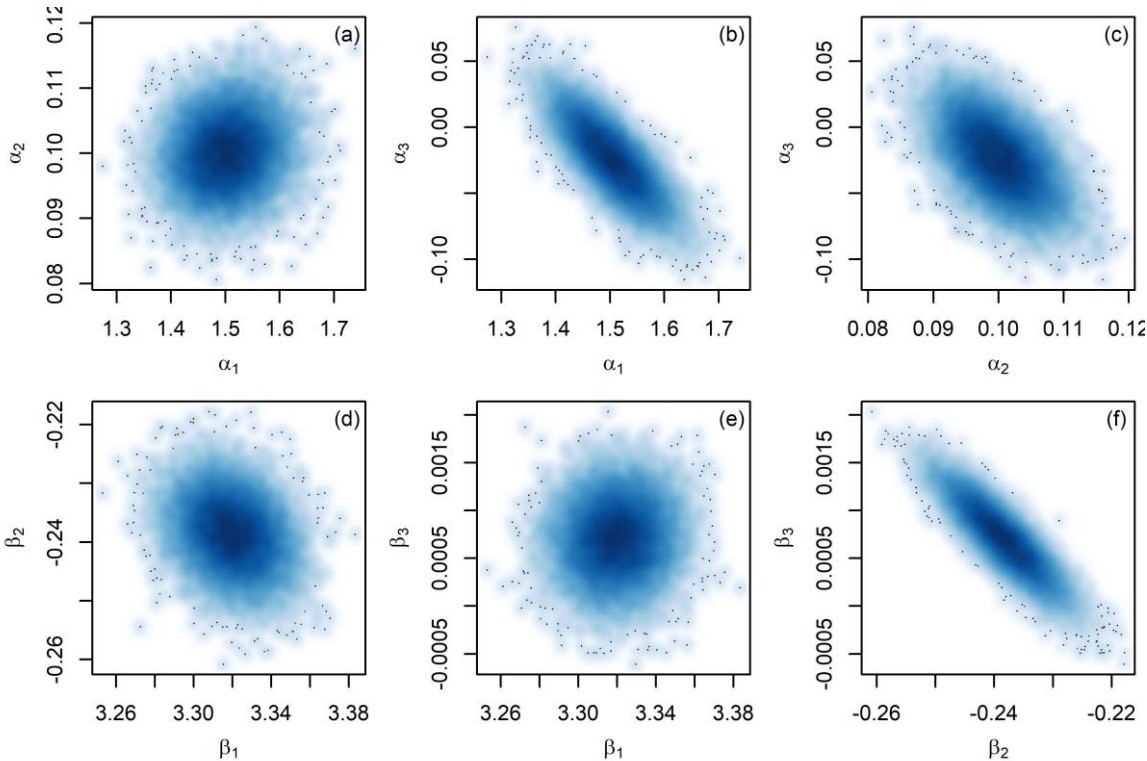

**Figure 6: Bivariate density plots of the posterior distributions for Mg/Ca (a-c) and δ¹⁸O (d-f) proxy model parameters.**





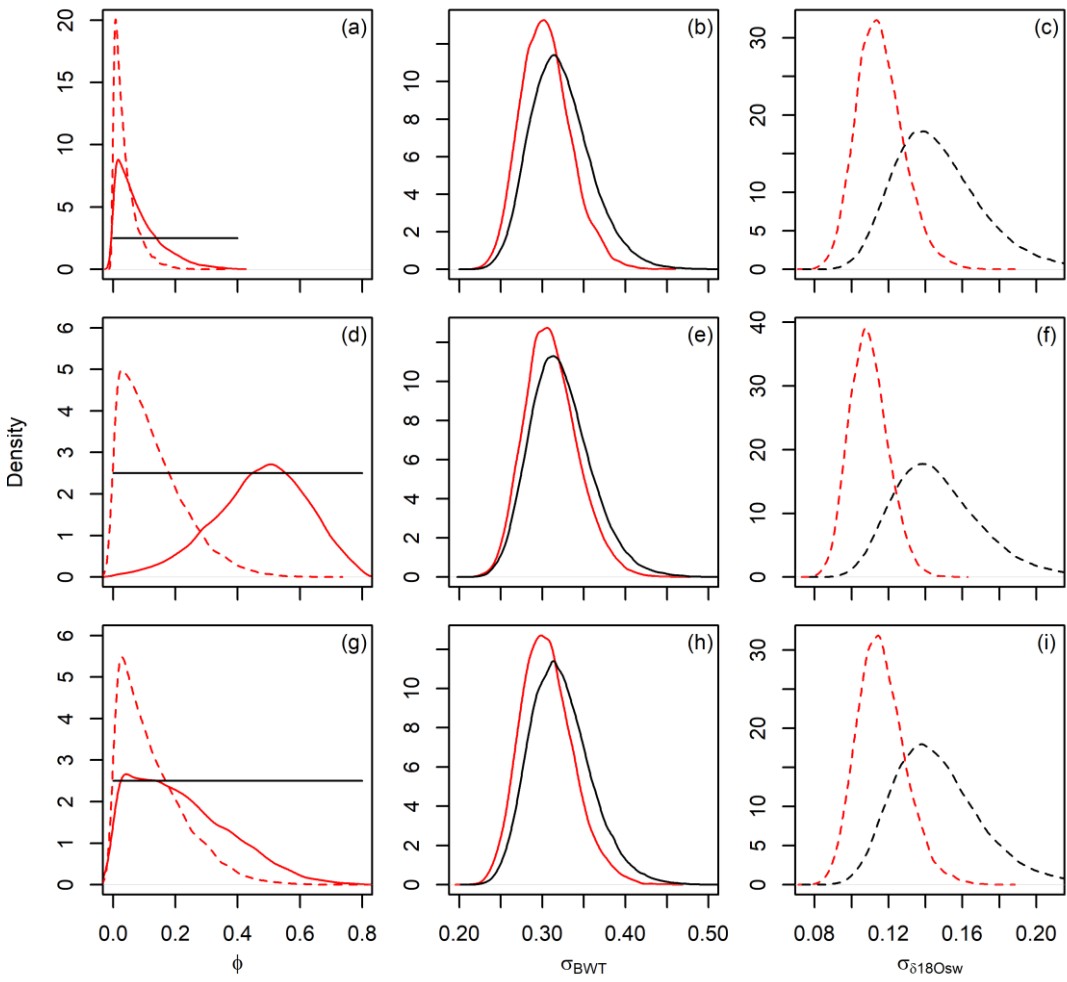

**Figure 7: Prior (black) and posterior (red) parameter distributions for bottom water temperature ($BWT$, solid) and seawater $\delta^{18}$O ($\delta^{18}O_{sw}$, dashed) time series models. (a-c) Site 806. (d-f) Site U1385. (g-i) Site 1123. (a, d, and g) Error autocorrelation (models for both variable used the same prior in a given analysis, shown here in solid black), (b, e, and h) standard deviation of $BWT$ error term, and (c, f, and i) standard deviation of $\delta^{18}O_{sw}$ error term.**





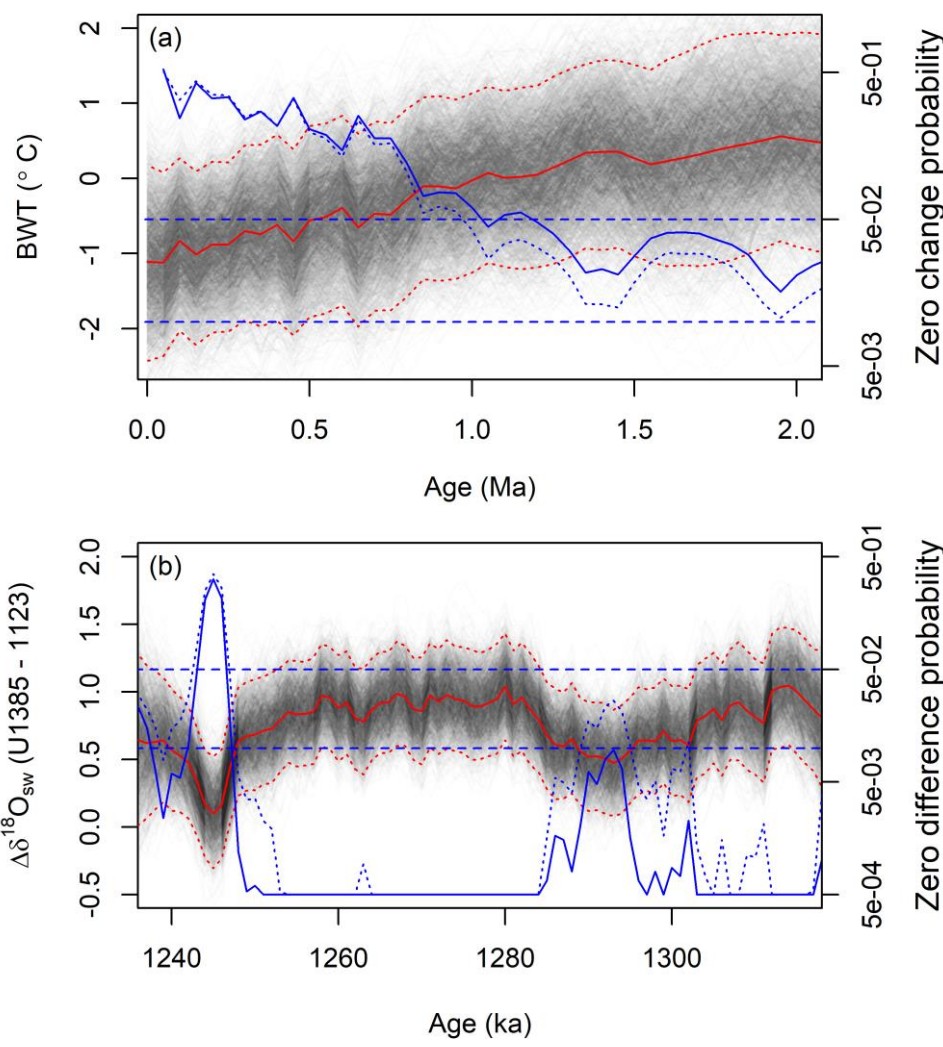

**Figure 8: Evaluating changes within and between environmental reconstructions using JPI output. (a) Site 809 bottom water temperature reconstruction from ~2 Ma to present, and probability of no significant change in temperature relative to modern. Grey and red lines show the BWT record. The blue solid and dotted lines show estimated probability of no change relative to modern, calculated based on change within (solid) or between (dotted) individual posterior samples. (b) Difference between site U1385 and 1123 seawater δ¹⁸O values within individual posterior samples, and probabilities of no significant difference between sites based on comparisons within (solid) or between (dotted) individual posterior samples. Blue dashed lines in both panels show 5% and 1% probability thresholds; all other symbols as in Fig. 2. See text for details.**





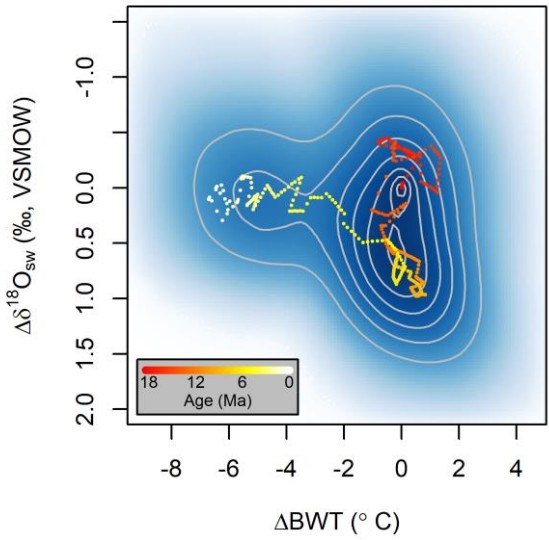

**Figure 9: Bivariate density plot of posterior values from the environmental time series models. All values are plotted as change relative to 18 Ma within an individual posterior sample. Dots show the median values from the posterior time series.**