# Peer review of "Joint inversion of proxy system models to reconstruct paleoenvironmental time series from heterogeneous data"

_Climate of the Past, 2018_

## Referee Comment (RC2) · Anonymous Referee #2 · 19 Apr 2019

Bowen et al. combine several proxy system models in the frame of a Bayesian hierarchical model to reconstruct seawater Mg/Ca, bottom water temperatures and surface water 18O based on Mg/Ca proxies and Mg/Ca and 18O measurements on foraminifera. This is an excellent manuscript and I recommend publication in Climate of the Past.

Major comment:

Parts of the methods section were difficult to assess because of missing references.

Did the authors develop proxy system models described in equations 2 and 3 or are these described elsewhere?

[Figure]

Page 4 line 30: How were these uncertainties determined?

Page 5 line 27: How is paleo-seawater Mg/Ca determined?

Page 4 line 30: How were bottom water temperature (BWT) uncertainties estimated?

Minor comments:

As far as I understood page 5 lines 25 – 32, proxy system model parameters are estimated based on observed (and inferred) BWT, surface water Mg/Ca and Mg/Ca of foraminifera. The posterior distributions of these parameters are then used as prior distributions when past surface water Mg/Ca and BWT are reconstructed.

The authors assume a paleo-seawater Mg/Ca of 1.5 when calibrating proxy system models. How do the authors get this value and how uncertain is it? How would including uncertainties affect parameter estimates?

Page 4: lines 28 and 29: some BWT values for calibration are based on 18O thermometry. Please explain this method (and add references). Is 18O thermometry based on eq 3? If yes, how were surface water 18O values determined and how do these values influence surface water 18O values reconstructed in this study?

Equation 2: Mg/Ca of foraminifera is modeled as a function of BWT and surface water Mg/Ca. However, credible intervals of alpha3 clearly include 0 indicative of weak (or absent) influence of surface water Mg/Ca on Mg/Ca of foraminifera, which might explain the results described page 8 line 5 (proxy data doesn't seem to inform this parameter either Fig 5c). Why is surface water Mg/Ca included in this proxy model given that it doesn't have a clear influence on Mg/Ca of foraminifera?

Equation 3: 18O of foraminifera is modeled as a function of 18O of surface water, BWT and BWT^2. However, credible intervals of beta3 (parameter relating BTW^2 and 18O) include 0 for Cibicioides as well as Uvigerina. Including BTW^2 in the model therefore needs additional justification. As the authors note in the discussion, posterior distributions of beta3 place even more weight on values close to 0 than the prior distribution.

---

## Author Response (AR1)

August 1, 2019

Helen McGregor
Editor, Climate of the Past

Dear Dr. McGregor:

We are pleased to resubmit our revised manuscript introducing the Joint Proxy Inversion method. We have addressed all reviewer comments in our manuscript and in the response text which follows this letter. Although none of the major findings, and few of the minor details of our results have changed, these updates have make the manuscript a much stronger contribution, and we thank you and the reviewers for their time and input.

The most substantial change is the use of a continuous-time formulation of the paleoenvironmental time series models, as suggested by reviewer 1. This has eliminated the need to adopt the (granted somewhat arbitrary) block interpolation method used to forward-model the proxy data in our initial submission, and makes the method more flexible overall. This all comes at the expense of increased computation time, and as we noted above produced no detectable changes in the fundamental results or interpretation of the records. However, we have updated the manuscript to present this formulation since it does in a fundamental sense represent an improvement over our previous approach.

As requested, we have released our GitHub repository through Zenodo, and now include a DOI pointing to this release as a persistent archive of the code and data.

Beyond that, additional details, clarification of the presentation and discussion, and a new supplementary figure showing the proxy calibration data and model posterior have been included. We have checked over the manuscript and materials, and hope that you agree that the contribution is now acceptable and ready for publication in *CoP*.

Sincerely,

Gabriel Bowen                                   gabe.bowen@utah.edu
Professor                                       (801)585-7925

Reviewer comments are shown here in plain text, our initial responses in italics, and the final response/revision in bold italics.

-Reviewer 1-

A Bayesian Hierarchical Model is used to reconstruct several environmental variables using some proxy variables, at three marine sites. The method isn't new e.g. Garreta et al. (2010), but the manuscript is useful as another example of this type of approach. The manuscript is unclear in places.

*We're glad the reviewer finds value in the paper. As acknowledged in the text and citations the approach here is not without precedent. We'll add the Garreta example, another interesting one using a much more complex (dynamic vegetation) process model to our reference list in the revision.*

***We now cite the Garreta paper as an example of previous applications in the introduction.***

In S2.3 (Section 2.3) Eq. 5 is a stationary discrete-time first-order autoregression (AR1) model for the random walk disturbances eY . According to S2.3, for BWT and d18Osw this AR1 model is run at a time step of 50 kyr (site 806) and 1 kyr (site 1123 and U1385). For Mg=Casw, the manuscript states "1 Myr time steps from 80 Ma to present", but what about for the higher-resolution sites (for Mg/Casw)? Also it is not clear what happens when the model time steps are different from the marine proxy time steps (which are irregular, S2.1 paragraph 2) - this point needs to be clarifed. It would be good to have a graphical depiction of the method (e.g. included in Fig. 1), with example time series (with clear time points) showing eY, Y, modelled proxy time series e.g. d18Of, and observed time series. Just show a portion of the time series, so that the time points for each time series are clear. # Further, instead of a discrete-time model, why not use a continuous-time model, which handles irregular time steps better than a discrete-time model. For example, a continuous-time time AR1 model is: (equation) For a continuous-time AR model, the parameters are not a function of the sampling intensity (Tomasson, 2015).

*We thank the reviewer for this suggestion, and in preparing our revision will explore the idea of using a continuous-time AR1 model. This would have clear benefits (including eliminating the need for interpolation, discussed below) and assuming it is feasible for the study systems we'll intended to adopt this tweak in the revision. We do not anticipate that the change will have a strong (or maybe even detectable) impact on the key outcomes. We will also work with the reviewer's idea of illustrating the time-series model properties through an addition to figure 1, which we like in theory and will do our best to implement without compromising the clarity/focus of the figure.*

***All simulations now use a continuous-time AR1 formulation, as suggested. This produces no material change in the results obtained, but does allow more robust treatment of irregularly spaced proxy time series (previously we used block interpolation to estimate values at proxy observation points). We note as an aside that using the continuous form does increase analysis time substantially as a result of the larger number of time steps to be evaluated, so our original block interpolation scheme might be preferable in some cases.***

***We have also added a visualization of the model framework to figure 1.***

S2.4 paragraph 3: "we conducted three different analyses ... the third inverting both records together." It's not clear what is meant by the latter phrase. An example of a discrete-time vector correlated random walk model is: (equation) … Exactly what model was used in "inverting both records

together", together with an explanation of why it would be mathematically different from "inverting data from each site independently" needs to be included in the manuscript text. Further there are vector continuous-time series models, which might be better to use for inverting multiple time series with irregular time intervals.

*The presentation here is somewhat ambiguous, and will resolve this in the revision. Our analysis remains agnostic of the correlation structure between the paleoenvironmental state variables at the two sites. They are modeled as independent time series, with no correlation term. We recognize that alternative models, such as that proposed by the reviewer, would allow incorporation of additional prior information and perhaps provide stronger process model constraints on the paleoenvironmental time series, acknowledging that they are likely not truly independent. However, the model proposed by the reviewer is just one step along a continuum of model forms that one could apply which would, at its end point, lead to a climate system model that expressed a full set of physics-based expectations for the relationship between the environmental state variables at the sites. While we acknowledge the potential value in such an analysis (which would basically become a data assimilation analysis), we are proposing and exploring a framework that lies at the other end of the continuum. Our goal is to offer a widely applicable and approachable framework in which practitioners who already routinely develop quantitative interpretations of their data without reference to any formal statistical framework or paleoenvironmental model can begin to adopt such without compromising the data-driven nature of their interpretations or having to frame them in the context of the complexity and structural assumptions of more complex paleoenvironmental models.*

**We now elaborate the form of this model and provide a few lines of context in the paragraph referenced by the reviewer.**

S3.2 paragraph 1: In statistics, the idea of smoothing (whether by frequentist or bayesian methods) stems from the idea that a time series = state variable + noise. Looking at Lear et al. (2015), the L15 reconstruction appears to have a higher variability simply because there was no smoothing employed. A better comparison here would be to create a reconstruction using both frequentist smoothing and bayesian smoothing methods, and then compare. The current comparison here seems a bit apples and oranges.

*Indeed, the crux of this comparison is smoothing, and based on the reviewer's feedback we propose to emphasize this more clearly with revisions to the language in this part of the discussion. We prefer not to present this as a comparison of Bayesian and frequentist smoothing techniques, however. The crux of our paper is not to enter into the Bayesian vs. frequentist discussion. Instead, we are trying to present an alternative to the reconstruction methods used nearly ubiquitously in the (pre-Holocene paleoclimate) community (and honestly most Holocene work), which do not embrace or consider concepts such as smoothing, multi-variate proxy models, or temporal autocorrelation of environmental timeseries. Our point in this section is that 1) smoothed reconstructions are a more realistic/honest expression of the information contained in proxy timeseries records, and that 2) the method demonstrated here offers an approach to optimize the properties of the smoothed reconstruction based on the data, rather than adopting an ad hoc approach (e.g., splines or running averages with arbitrarily specified parameters) as is commonly done if and when smoothing is conducted. The comparison is apples to oranges, but we think also of value.*

**We have edited these paragraphs to better emphasize the concept of smoothing as relevant here and note that other approaches to smoothing have and can be used.**

S3.2 paragraph 3: So if d18Osw and BWT are generated at 1 kyr time steps, and the sampling resolution of d18Of is between 1 per 110 and 1 per 1700 years, do you generate the model time series first, at 1 ka steps, and then use Eq. 5 to "integrate" to the proxy time points (if necessary)? How is that integration done?

*In our original analysis, values for proxy time points are obtained using a 'nearest neighbor' approach, i.e. the value at the nearest proxy time series point is used. We will clarify and discuss/justify this in the revision if we end up maintaining a discrete time series model approach, or if we adopt the continuous AR1 model this will become a moot point.*

**This issue is now moot as a continuous-time AR model is used to estimate at each proxy time point.**

S3.2 paragraph 4: The following sentence could be worded better: "Moreover, because temporal autocorrelation of the environmental variables is considered ...". I think you are trying to say its both the autocorrelation (in the environmental states) and sample density which make the credible intervals what they are. In the next sentence, can you explain mathematically what is meant by "the strength of the proxy constraints"?

*Yes, we can work on rewording/elaborating to clarify as requested.*

**This text has been re-written to clarify the point raised in the reviewer comment.**

S3.3 paragraph 2 ("These refinements reflect ...") After 800 ka, perhaps the higher proxy model variance is suggesting the environmental model is missing something? For example, what would be the effect of adding a stochastic periodic component to the process model to capture the 100 ka cycle after 800 ka?

*Absolutely, this conclusion is essentially what we were suggesting here. Adapting the process model would be a good, perhaps more appropriate, alternative to adapting the data model across the 800 kya boundary, and we propose to explore this alternative in preparing our revision. This will depend a bit on whether the sampling resolution of the site 806 data is adequate to constrain the sub-100 kyr variability in this interval, in which case it makes sense to treat it as 'signal' (i.e. in the process model) or 'noise' (i.e. as done, in the error term of the data model).*

**We experimented with this a bit but given that our analysis of this dataset focuses on Myr-scale trends we have opted to retain the simpler time series model parameterization in the revised manuscript. We now explain the rationale for this decision in the methods (S2.2), however, and note that using a more complex environmental model is an alterative that would be preferable under some other circumstances.**

S3.3 paragraph 3 The phrase "double-count uncertainty associated with correlated parameters" is not an elegant mathematical explanation.

*With all respect, and acknowledging the suggestion, we are not writing this for mathematicians but rather for paleoclimate practitioners. Here we are attempting to provide a common-language explanation of some of the contrasts between the proposed approach and common practice. This phrase may be somewhat imprecise, but we think it makes the point in a way that most readers will grasp it.*

***We have changed the language here and now use the more technical term "inflate".***

It's unclear exactly how the dotted blue line in Fig. 8a is calculated. Explain. Also statistical tests don't always need to assume independence, because there are ways of accounting for autocorrelation in a statistical test.

"The net result in this case ... some 100-200 kyr earlier using the traditional approach": would this sentence be true if autocorrelation was taken into account in the traditional approach. I'm looking for a fair comparison here.

Also for the solid blue line in Fig 8a - give details of its calculation.

*We will happily elaborate/be more specific on the calculation of the 'traditional' analyses presented in the figure. In the original draft we had erred on the side of brevity in an attempt to interrupt the flow during the latter part of the manuscript. We see how this compromises the clarity of the analysis, however, and will revise to ensure the calculations are described in enough detail to be reproducible from the text alone (i.e. not requiring reference to the data analysis code, which is already publically available and fully documents the details). With respect to autocorrelation – indeed this is the crux of the difference noted in figure 8a. We will try to make this clearer/provide greater emphasis in the revision. Akin to the comment above on smoothing, our point here is that the JPI framework integrates explicit treatment of time series autocorrelation, ensuring that data interpretations developed from the method reflect a robust consideration of such, unlike many analyses presented in the literature. There are other ways of achieving this, of course, and we'll be sure to better make that point in our revision.*

***We have added the details of how each metric of change is calculated to the figure 8 caption. We have also reframed the discussion of the first comparison (Fig. 8a) to provide examples of how such analyses have been accomplished previously, emphasize the value of integrating the estimation of autocorrelation within the JPI analysis, and focus more on what learned from internal comparison rather than benchmarking against a 'traditional' method external to our work.***

p4 L13: "sampling resolution between 1 per 110 and 1 per 1700" years. Clarify for d18O, Mg/Ca, or both?

*This summarizes across both proxies, and we will clarify in the revised text.*

***Done***

p5 L11-12: The Evans et al. (2013) terminology includes "sensor models", "archive models", and "observation models". Clarify which of your equations relate to which type of Evans's models?

*I personally have struggled with this, as I don't think there is a 1:1 mapping in this case. Part of the issue is that the Evans et al conception includes a strong focus on the processes that integrate proxy responses in a biological or sedimentary medium that accumulates over time (e.g., sediment stack, incremental growth structure; archive model) and how those integration processes are sampled (observation model). These are not explicitly treated here, or in many proxy interpretations, and in some lower-resolution deep time studies may be less critical than in much of the higher-resolution shallow-time work (I'll note, however, that I'm not sure I actually believe this…it is a frontier area*

*and there are now a handful of really interesting avenues being pursued, e.g., with respect to processes such as seasonal sampling of different proxy archives and the impact of sedimentary architecture and allogenic processes on signal integration/preservation). At any rate, what we have here, in my interpretation, is primarily a sensor model, which also embeds some aspect of what would appear in archive and observations models in the proxy model error term. We will state this (more concisely that I have here!) in the revision.*

**We have expanded this sentence to elaborate how the other component models of Evans et al. map to our equations.**

p5 L17: "age estimate and uncertainty" Ambiguous wording, because as is it reads "age estimate and age uncertainty".

*We will reword as suggested.*

**Wording has been changed to eliminate ambiguity.**

Eq. 2 and 3: For clarity, can you make all the "functions" with round brackets e.g. BWT(tMgCaf [i]). Change the outer brackets too i.e. {}. Keep square brackets for distributions e.g. N[], as you have done.

*We will reformat the equations as suggested.*

**Done.**

Eq. 5 Say what Y can be e.g. Y (t) can be MgCasw(t) or d18Osw(t) or BWT(t).

*We will elaborate as suggested.*

**Done.**

p8 L2: Clarify the phrase "stiff" time series behaviour (give a reference if possible)

*We will do some literature research to see if we can come up with a more formal way to express this result...we were trying here to colloquially express the condition in which error variance is small and error autocorrelation large, as for the Mg/Ca_sw, which leads to long burn-in times using most methods for generating MCMC step proposals.*

**Upon further investigation, we have removed this extraneous 'common-language' term from the description of the Mg/Ca_sw time series.**

p12 L8: "Across all scales": Across all sites?

*We intended 'across all timescales', but the suggested 'across all sites' would probably be clearer and will be adopted in the revision.*

**Done.**

Additional figures showing the calibration datasets, with individual draws from the posterior distribution, should be included. These could go in the manuscript or supplementary material.

*We can easily add these to the SI in the revision, and are happy to do so.*

**New figure S4 shows examples of the calibration relationships as requested.**

Fig. 5: Which inversion did these distributions come from e.g. site 806? (include in caption)

Fig 6: same comment as Fig. 5.

*We will clarify that these are from the 806 analysis and also indicate the taxon to which they apply in the revised figure legends.*

**We have added this information to the figure captions.**

Fig. 7: The prior distributions in (d) and (g) don't integrate to 1.0 e.g. 2.5 x 0.8 = 2.0. I can't tell if all the other distributions integrate to one or not.

*Thanks for catching this…it is a plotting error (we carried over the y-axis value of 2.5 appropriate to the prior in panel a) which we will correct in the revision.*

**This error has been fixed.**

Figure 9: There is a positive relationship between DBWT and Dd18Osw in the two Miocene states (mentioned in the last sentence in S3.4). I think adding some straight lines to mark this, and not inverting the y-axis here would help the reader.

*We will certainly un-invert the y axis as suggested (appropriate here since we are plotting d18O of seawater and not carbonate). We will also explore ways of adding lines that represent the correlation in posterior values from different states.*

**We have flipped the y-axis. We have not added lines showing the positive DBWT/Dd18Osw correlation during the two Miocene states since this feature is only weakly expressed in the revised analysis.**

-Reviewer 2-

Parts of the methods section were difficult to assess because of missing references.

*We assume that the following questions point to specific cases where additional citation would be helpful, and address how and where this will be resolved in our responses below.*

**See below**

Did the authors develop proxy system models described in equations 2 and 3 or are these described elsewhere?

*These equations represent 'standard' widely-used forms used to describe the temperature sensitivity of foraminiferal calcite Mg/Ca and $\delta^{18}O$ values in the literature. In our revision we will make this clear and cite some of the literature in which these forms have been previously proposed and used. We believe this will also help address some of the reviewer's later questions about the rationale for including some of the terms in these proxy model equations (we're adopting/testing equation forms based on precedent in the community).*

**We have elaborated and provided references in this section describing the precedent for these equations.**

Page 4 line 30: How were these uncertainties determined?

*These are approximations derived from the original data sources, we will clarify this point and add citations in the revision.*

**We have reworded this section to clarify the source of the (ballpark) uncertainty estimates.**

Page 5 line 27: How is paleo-seawater Mg/Ca determined?

*The value given here in the text here are simply first-pass estimates used in developing the priors on the foram proxy model parameters. For the non-modern (Paleocene-Eocene) samples we use a value of 1.5 mol/mol based on prior work of Lear et al. (2015). We will clarify this and add the citation to this sentence in the revision.*

**We now cite the source of this estimate in the referenced sentence.**

Page 4 line 30: How were bottom water temperature (BWT) uncertainties estimated?

*Answered above.*

**See above.**

As far as I understood page 5 lines 25 – 32, proxy system model parameters are estimated based on observed (and inferred) BWT, surface water Mg/Ca and Mg/Ca of foraminifera. The posterior distributions of these parameters are then used as prior distributions when past surface water Mg/Ca and BWT are reconstructed.

*This is almost but not quite correct. In our framework, posterior distributions for all parameters (including the proxy model parameters and the paleoenvironmental parameters) are found together. In other words, rather than first estimating the posterior of the proxy model parameters, then applying them to estimate the posterior distributions of the paleoenvironmental (process) model parameters, we simulate both sets together. To envision one implication, imagine that the 'true' value of one of the proxy model parameters (let's say temperature sensitivity of foraminiferal Mg/Ca) was actually a bit higher than the average estimate. Given that, the most likely paleo-environmental temperature time series would be shifted relative to the 'mean' estimate, also. By solving the full system simultaneously the joint distribution of posterior parameters captures these trade-offs and can be analyzed in new ways (e.g., see some of the derived analyses later in the paper).*

***We have added a statement to this section clarifying that the initial regressions are used to develop prior distributions for the proxy model parameters.***

The authors assume a paleo-seawater Mg/Ca of 1.5 when calibrating proxy system models. How do the authors get this value and how uncertain is it? How would including uncertainties affect parameter estimates?

*As mentioned above, this value was only used in estimating the prior distribution for the Mg/Ca model parameters (and only for one species). The parameter values contained in the posterior distribution are the result of MCMC sampling of the entire model system. We have replicated the analysis using a variety of prior assumptions for the foraminiferal Mg/Ca proxy model and find little sensitivity in the posterior distributions (not shown). We will try to make this logic clearer in the revision.*

***Although we don't present a formal analysis of the impact of this assumption in the manuscript, our other results (e.g., figures 2 and 5) show that 1) the Mg/Ca value used in the estimation of proxy model priors is similar to that inferred in the full JPI analysis and should not bias the result, and 2) the priors we obtained in this way are consistent with the posterior estimates from the full inversion despite the fact that the full JPI provides additional constraints on these parameters…in other words there is a strong suggestion here (figure 5) that the prior is not strongly affecting the results obtained in the full analysis.***

Page 4: lines 28 and 29: some BWT values for calibration are based on 18O thermometry. Please explain this method (and add references). Is 18O thermometry based on eq 3? If yes, how were surface water 18O values determined and how do these values influence surface water 18O values reconstructed in this study?

*These values were only used for calibrations including data from the early Paleogene, when the globe was essentially ice-free. The BWT estimates are thus based on 'standard' assumptions for the $\delta^{18}O$ of the ice-free ocean. The actual values used are from Lear et al. (2015), and we will add this citation to the sentence for clarity.*

***The text states the ice-free assumption and now cites the source of the estimates.***

Equation 2: Mg/Ca of foraminifera is modeled as a function of BWT and surface water Mg/Ca. However, credible intervals of alpha3 clearly include 0 indicative of weak (or absent) influence of surface water Mg/Ca on Mg/Ca of foraminifera, which might explain the results described page 8 line 5 (proxy data doesn't seem to inform this parameter either Fig 5c). Why is surface water Mg/Ca included in this proxy model given that it doesn't have a clear influence on Mg/Ca of foraminifera? # Equation 3: 18O of foraminifera is modeled as a function of 18O of surface water, BWT and BWT^2. However, credible intervals of beta3 (parameter relating BTW^2 and 18O) include 0 for Cibicioides as well as Uvigerina. Including BTW^2 in the model therefore needs additional justification. As the authors note in the discussion, posterior distributions of beta3 place even more weight on values close to 0 than the prior distribution.

*In both cases our approach was to adopt the model forms commonly in current use within the paleoclimate community (ref. our response to the earlier question). These forms have been adopted, usually based on empirical relationships rather than fundamental considerations, and widely used in previous studies, and we chose them for consistency and comparability with prior work. However, as*

*noted by the reviewer, in some cases the results of our analysis suggest that one or more model parameters are not or only weakly informative. This result has been noted before in studies that have used traditional statistical approaches to calibrate model equations for these systems. The sensitivity to these parameters seems to vary among species, however, so that in most studies the full form of the equations (all terms) are considered so that the same form can be used for all species. Although we propose to continue to use the 'canonical' forms in our revision, we will better emphasize and elaborate on the result that our analyses do not support sensitivity to some of these model terms, which may 1) suggest that, for these species, a simpler proxy model is appropriate, and 2) slightly inflate uncertainty estimates when these terms are included.*

***We have added a statement in the discussion noting the lack of sensitivity to these parameters and indicating why we retain them in our analysis.***

[revised manuscript text omitted]

---

## Referee Report (RR1)

**Review**

Manuscript: Joint inversion of proxy system models to reconstruct paleoenvironmental time series of heterogeneous data
Authors: G. J. Bowen, B. Fisher-Femal, G.-J. Reichart, A. Sluijs and C. H. Lear
Manuscript Number: CP-2018-178

**General comments**

The authors have generally answered my comments, including implementing a continuous-time model rather than a discrete-time model. In their revisions, there are some new points needing clarification.

**Specific comments**

1) *Section 2.3 Environmental models*
Now that a continuous time series (cts) model is being used (Eq. 5), it's unclear why the process model is simulated with a "regularly-spaced base series" (paragraph under Eq. 5), rather than just sampling the cts model at the observed (data) time points. Sampling the cts model at only the data time points preserves the uncertainty caused by changes in temporal resolution. (The link between temporal resolution and confidence intervals is mentioned at the end of Section 3.2 in the manuscript.) What is unclear in the manuscript is whether $\Delta t$ (in Eq. 5) is:

a) $t_i - t_{\text{last regularly spaced point}}$, or
b) $t_i - t_{i-1}$ i.e. the time difference between the observed time points.

If $a < b$ then the variance of the smoother may be artificially smaller.

Also a reference to a statistical text that provides the background theory about continuous time correlated random walk models (i.e. the origins of Eq. 5) should be included.

2) *Section 2.4 Model inversion*
"we conducted three different analyses, the first two inverting data from each site independently and third inverting both records together". Although the authors have given a better description of the latter (in their subsequent sentences), the "why" remains unclear. Is there any expectation here that inverting both records together should for some reason be different than inverting both records independently? Explain why or why not.

**Technical corrections**

(page and Line numbers refer to the clean revised manuscript)

Fig. 1: In Figure 1b and caption, the symbol $\varepsilon$ is used, whereas $\epsilon$ is used in the manuscript text. Please clarify.

p5 L22 "$t_{swp}$ and $\sigma_{swp}$ are the estimated age and uncertainty" is still ambiguous. Please add an adjective before "uncertainty", because $\sigma_{swp}$ is not age uncertainty (and that's how it reads).

p9 L23 "original authors" I'd clarify this by saying "Both reconstructions are similar in nature to the reconstructions provided by ..." and cite the papers.

p10 L7 "values of these parameters" change to "values of these variables". I think the word "parameter" should be reserved e.g. for the coefficients of a statistical model.

p10 Heading 3.2 "Time series properties" vs 3.3 "Model properties" In the opening paragraph of these sections, please give a definition of "time series property" and "model property".

p12 L32-L34 The median value of the autocorrelation ($\sim$0.9) is given for site 806, but please give the median value of the autocorrelation value (in brackets) whenever you qualitatively state "strongest autocorrelation" or "much lower posterior" in this paragraph. Also, the autocorrelation posterior density (Fig. 7d, solid line) for BWT seems to suggest a weak autocorrelation, which doesn't match the sentence on p13 L1 "the data strongly support highly coherent high-amplitude cyclic variation in BWT". Given such a pattern, I would have expected a higher autocorrelation.

p13 L9: "providing ... supporting" or "provide ... (thus) supporting".

p13 L13 "ad hoc methodological choices" Just a comment. But there are other issues with dynamic linear models, including choice of the components of the process model, and the lack of model cross-validation.

p13 L17 How much is a "slightly higher probability of a significant change"? (include numerical values in brackets). Is the probability of *no* change from Modern higher or lower at 0 Ma for which method?

Fig. 8a caption "Site 806" not "809"
Fig. 8b The red line is not referred to in the 8b caption (one can guess that it corresponds to the left y-axis).
Is the phrase "all other symbols as in Fig. 2" necessary?

---

## Author Response (AR2)

September 28, 2019

Dr. Helen McGregor
Handling Editor
Climate of the Past

Dear Editor McGregor:

We are submitted a further revised version of our manuscript, having considered and incorporated the feedback provided in a re-review by one of the original reviewers. These were all issues of clarification and elaboration, and in each case we have made minor changes to the manuscript text that have improved the presentation of our work. No new analyses have been conducted (with the exception of a minor 'experiment' conducted to demonstrate a point that is included only in the response document below) nor have any results or interpretations changed relative to the prior version.

Please let us know if you need any additional information or clarification in considering this revision, and thanks for your effort handling our submission.

Sincerely,

Gabriel Bowen                                          gabe.bowen@utah.edu
Professor                                              (801)585-7925

Response to reviewer comments (comments in italics, response in bold).

*1) Section 2.3 Environmental models*

*Now that a continuous time series (cts) model is being used (Eq. 5), it's unclear why the process model is simulated with a \regularly-spaced base series" (paragraph under Eq. 5), rather than just sampling the cts model at the observed (data) time points. Sampling the cts model at only the data time points preserves the uncertainty caused by changes in temporal resolution. (The link between temporal resolution and confidence intervals is mentioned at the end of Section 3.2 in the manuscript.) What is unclear in the manuscript is whether Dt (in Eq. 5) is:*

*a) $t_i - t_{last\ regularly\ spaced\ point}$, or*

*b) $t_i - t_{i-1}$ i.e. the time difference between the observed time points.*

*If a < b then the variance of the smoother may be artificially smaller.*

*Also a reference to a statistical text that provides the background theory about continuous time correlated random walk models (i.e. the origins of Eq. 5) should be included.*

**Three issues are raised here:**

**1.1) How is the time series sampled and why? The cts model is sampled at all base and proxy observation time steps, and we have modified the text to make this more explicit: "Each variable is modeled on a time series composed of a regularly-spaced base series appropriate to the record duration and resolution plus all proxy sample ages, with Δ*t* representing the time shift between all adjacent base and proxy ages". The goal here is illustrate how information from the irregularly- and discretely-sampled proxy observations propagates to constrain the paleoenvironmental state estimates \*between\* observation points, which is possible specifically because of the inclusion of the time-series process model in the JPI method. This is also critical, for example, for supporting cross-site comparisons where the proxy series are differently sampled at different sites (e.g., Fig. 8b).**

**1.2) Does this sampling strategy bias the variance of the smoother? As noted in the reviewer's comments on the original submission, the behavior of a correctly-posed cts model should be independent of step size, and this is the case here. The variance of the error term in our cts model scales with step size, and as a result the outcome is not sensitive to the choice of time step size. To demonstrate we have re-run our analysis for a 40kyr segment of the U1385 dataset, first using the time series sampling strategy used in our manuscript (1-kyr base series plus all proxy observation time-points, red) and then using a time series consisting only of the proxy observation time-points (blue). Both inversions were run to an equal number of posterior samples and to convergence. The mean and 95% CIs are shown below. The CI bounds are nearly identical, and in fact in the less densely sampled parts of the record the CIs for the method used in the manuscript are slightly broader. So in response to the reviewer's assertion that the time series sampling strategy used here might artificially bias the smoothing of the record, we assert that no, it does not.**

[Figure]

**1.3) Provide reference for the cts model. We have provided a reference and some additional explanation to accompany the model. The starting point is the classical Ornstein-Uhlenbeck process, and here we have modified the variance term to reproduce the variance scaling of the discrete-time AR1 model (explained in text).**

*2) Section 2.4 Model inversion*

*"we conducted three different analyses, the first two inverting data from each site independently and third inverting both records together". Although the authors have given a better description of the latter (in their subsequent sentences), the "why" remains unclear. Is there any expectation here that inverting both records together should for some reason be different than inverting both records independently? Explain why or why not.*

**Yes, there is an expectation of difference related to the use of a common proxy model (aka data model) for both sites in the joint analysis, which imposes a co-dependence on paleoenvironmental timeseries at the two sites within individual posterior samples. We have added two sentences to this section to clarify/expound on this point: "The use of these common data models constitutes the primary difference between the two analyses, in that individual posterior samples from the joint analysis include paleoenvironmental time series at both sites that are consistent with a single set of data model parameters. The implicit assumption behind this approach is that the proxy calibration is imperfectly known but that 'correct' calibration, if known, would be the same at the two sites."**

*Technical corrections*

*Fig. 1: In Figure 1b and caption, the symbol $\varepsilon$ is used, whereas $\epsilon$ is used in the manuscript text. Please clarify.*

**Fixed.**

*p5 L22 "tswp and σswp are the estimated age and uncertainty" is still ambiguous. Please add an adjective before "uncertainty", because σswp is not age uncertainty (and that's how it reads).*

**Fixed.**

*p9 L23 "original authors" I'd clarify this by saying "Both reconstructions are similar in nature to the reconstructions provided by ..." and cite the papers.*

**Done.**

*p10 L7 "values of these parameters" change to "values of these variables". I think the word "parameter" should be reserved e.g. for the coeffcients of a statistical model.*

**Agreed, and changed.**

*p10 Heading 3.2 "Time series properties" vs 3.3 "Model properties" In the opening paragraph of these sections, please give a definition of "time series property" and "model property".*

**Done. We have added an introductory sentence to each of these sections explaining what is intended by the section heading.**

*p12 L32-L34 The median value of the autocorrelation (~0.9) is given for site 806, but please give the median value of the autocorrelation value (in brackets) whenever you qualitatively state "strongest autocorrelation" or "much lower posterior" in this paragraph. Also, the autocorrelation posterior density (Fig. 7d, solid line) for BWT seems to suggest a weak autocorrelation, which doesn't match the sentence on p13 L1 "the data strongly support highly coherent high-amplitude cyclic variation in BWT". Given such a pattern, I would have expected a higher autocorrelation.*

**We have added quantitative values to accompany all of the qualitative statements in this paragraph. We have also revised and further elaborated on the value estimated for BWT at site U1385. Although it is the highest error autocorrelation value estimated for the Pleistocene records, it's true that it is a relatively low value, as noted by the reviewer. This is driven by relatively abrupt shifts in the proxy data that occur within a few parts of the record, and we now elaborate this point in the text: "Among the Pleistocene analyses, the strongest error autocorrelation is inferred for BWT at site U1385 (mean = 0.12). There, the data suggest coherent cyclic variation in BWT across two glacial cycles, consistent with stronger error autocorrelation, but several more abrupt, short-term shifts are also implied (e.g., at ~1.31 Ma) and likely reduce the posterior estimate of autocorrelation across the record as a whole."**

*p13 L9: "providing ... supporting" or "provide ... (thus) supporting".*

**We have reviewed this sentence and improved the wording, which now reads "…provide distributions for the environmental variables that support testing…"**

*p13 L13 "ad hoc methodological choices" Just a comment. But there are other issues with dynamic linear models, including choice of the components of the process model, and the lack of model cross-validation.*

**Fair enough, we have tempered the wording here (removing "ad hoc") and specified that we are talking about advantages related to the specific model components introduced in the first part of the sentence.**

*p13 L17 How much is a "slightly higher probability of a significant change"? (include numerical values in brackets). Is the probability of \*no\* change from Modern higher or lower at 0 Ma for which method?*

**We have added quantification as requested. The methods cannot be directly compared at 0 Ma given that the within-sample method is based on the distribution of differences between t and 0 Ma, which by definition is U[1,1] at t = 0 Ma.**

*Fig. 8a caption "Site 806" not "809"*

**Fixed, thanks.**

*Fig. 8b The red line is not referred to in the 8b caption (one can guess that it corresponds to the left y-axis).*

*Is the phrase "all other symbols as in Fig. 2" necessary?*

**We have added a description of the red lines and removed the reference to figure 2, which we agree is not necessary now.**

[revised manuscript text omitted]